# Synthesis, Photochromic and Luminescent Properties of Ammonium Salts of Spiropyrans

**DOI:** 10.3390/molecules27238492

**Published:** 2022-12-02

**Authors:** Artur A. Khuzin, Dim I. Galimov, Artur R. Tulyabaev, Liliya L. Khuzina

**Affiliations:** Institute of Petrochemistry and Catalysis, Ufa Federal Research Center of the Russian Academy of Sciences, 141 Oktyabrya Prospect, 450075 Ufa, Russia

**Keywords:** spiropyran, merocyanine, photochromism, spectral and kinetic properties, luminescent properties, photodegradation

## Abstract

New salts of photochromic indoline spiropyrans capable of reversibly responding to UV radiation were synthesized to develop light-controlled materials. Photoinduced reactions of the synthesized compounds were studied using absorption and luminescence spectroscopies, and the quantum yields of photoisomerization and other spectral and kinetic characteristics were measured. It was shown that the light sensitivity and photostability of the synthesized compounds are considerably influenced by the length of the spacer between the indole and ammonium nitrogen atoms.

## 1. Introduction

Spiropyrans form one of the most interesting classes of organic photochromic compounds capable of reversible isomerization under the action of external stimuli. The optical and other physicochemical properties of spiropyran isomers considerably differ; therefore, these photochromes can be used as sensors [1,2,3,4,5,6], optoelectronic and holographic devices [7,8], memory cells [9,10,11], etc. Moreover, an obvious advantage of spiropyrans over other classes of photochromic compounds is the relative ease of their synthesis and structural modification. Finally, modification of the spiropyran structure by introduction of functional groups offers broad opportunities for the targeted synthesis of new photochromes with a wide variation of the spectral and kinetic properties [12,13,14].

In addition, it is known that ionic liquids are capable of triggering apoptosis in cells via the mitochondrial pathway. Such molecules can be incorporated into membranes of eukaryotic cells and disrupt their integrity to cause their death due to long side alkyl chains in their structure [15,16,17].

Merocyanines obtained via photochemical isomerization are such ionic compounds. At the same time, the luminescent properties of the open forms of these molecules will not only allow the visualization of the pathway itself and the accumulation of these compounds in specific organelles of target cells, but they will also allow one to study the mechanism of cytotoxic activity in different cell lines. This would open up prospects of obtaining anticancer drugs based on them for the treatment of human oncological diseases due to the solubility of these compounds in water. The only work which describes photoswitchable cytotoxicity against Hek293 cancer cells using spiropyran as an example serves as the feasibility of this approach [18].

## 2. Results and Discussion

### 2.1. Chemistry

Here we continued these studies by synthesizing new indoline spiropyrans with the aim of expanding the scope of applicability of spirophotochromic compounds, obtaining new light-controlled materials with a variety of properties, and studying the effect of the length of the spacer between the indole and ammonium nitrogen atoms on the spectral and photochromic properties. The presence of the formyl group in the molecule gives hope that the obtained compounds would exhibit bright luminescence [19].

Spiropyrans were synthesized using methods reported in the literature [20,21] according to Figure 1:

The structures of compounds **22**–**26** were determined using ^1^H and ^13^C NMR spectroscopy and high-resolution mass spectrometry (HRMS, see Appendix A).

In the ^1^H NMR spectrum of compound **22** in CDCl3, the signal positions and integrated intensities, as well as the spin–spin coupling constants, are in line with the presented structure. For example, two singlets at 1.18 and 1.30 ppm correspond to methyl protons at the indole nitrogen atom. The singlet with a chemical shift at 2.52 ppm corresponds to protons of two methyl groups at the ammonium nitrogen. In the case of compound **17**, the singlet is shifted up-field and is observed at 2.21 ppm. The signal indicating the presence of a spirocyclic structure is a doublet in the aromatic region at 5.93 ppm with a spin–spin coupling constant of 10.3 Hz, which corresponds to the CH group at the spiro atom in the pyran moiety. The formyl proton signal is manifested at 9.77 ppm (see Appendix A).

In the ^13^C NMR spectrum of **22** in CDCl3, the number of signals is equal to the number of carbon atoms in the molecule. The characteristic signal of the spirocyclic carbon atom is manifested at 106.19 ppm and is correlated with the proton signal of the gem–methyl groups and the C3′ and C4′ proton signals in the ^1^H–^13^C HMBC spectrum. The formyl carbon signal is detected at 190.50 ppm (see Appendix A).

### 2.2. UV–Vis and Luminescent Studies

Considering the sensitivity of spiropyrans to a broad range of external stimuli [5,18,19,20,21,22,23,24,25,26,27,28,29], the photoinduced reactions of the synthesized salts **22**–**26** in tetrahydrofuran (THF) were studied using absorption spectroscopy. THF was chosen as the solvent due to its moderate polarity, high dissolving capacity, and inertness to the spiropyran salts.

Figure 1 shows the absorption spectra typical of spiropyrans **22**–**26** without and under UV irradiation, taking compound **22** as an example. The measured spectra exhibit four most characteristic bands, three being in the UV range with peaks at 266–267, 324–327, and 374–377 nm, and one being in the visible range with a maximum at 599–604 nm (Figure 1 and see Appendix A). The two first-mentioned bands are known to correspond to spiropyran molecules in the closed-ring form (Figure 1, curve 1), while the longer wavelength bands are attributable to absorption of molecules in the open merocyanine form (Figure 1, curve 2). The strong absorption of photochromic compounds **22**–**26** in the orange spectral range (599–604 nm) is responsible for the deep blue color of their solutions (Figure 2).

Upon UV irradiation, the intensities of bands at 266–267 and 324–327 nm somewhat decrease, while the intensities of the 374–377 and 599–604 nm bands considerably increase (Figure 1, spectra 2–10). These changes unambiguously indicate that UV irradiation induces photochromic transformations of spiropyrans in the adsorption layer to give the open merocyanine form of the molecules. The observed changes are reversible: molecules **22**–**26** cyclize again in the dark (dark bleaching) or under the action of visible light (photobleaching). This switching cycle can be repeated more than 10 times. The complete photoinduced transition of spiropyrans **22**–**26** (10^−4^ M in THF) from closed to open form takes 11–18 s. The reverse transition is characterized by a longer duration: 180–190 s with dark discoloration, and 100–170 s with visible light irradiation.

Analysis of the absorption spectra of compound **22** showed that the absorption bands with maxima at 374–377 and 599–604 nm are irregularly broadened and, as the period of spontaneous dark bleaching increases, the long wavelength absorption bands not only decrease in intensity, but also shift to shorter wavelengths. The resolution of experimental spectra into Gaussian bands (see Appendix A) demonstrates that the absorption spectra of merocyanine forms are complex. This attests to the presence of *cis*- and *trans*-isomers of open-ring spiropyrans (Figure 3), which are characterized by different stabilities and different lifetimes of the metastable state.

It can be seen from the data of Table 1 that the positions of absorption bands of compounds **22**–**26** remain almost invariable in both spirocyclic and merocyanine forms, i.e., they do not depend on the number of methylene groups in the spacer chain. This attests to the absence of electronic interactions between the N–amine moiety and the π-conjugated system in the electronic ground state of the molecule.

Study of the relaxation kinetics of compounds **22**–**26** without irradiation or on exposure to visible light showed that the kinetic curves are exponential (Figure 4A); their linearization in the ln(I)–t(s) coordinates gives linear plots for the initial stage (Figure 4B). This fact indicates that the reverse transition is a first-order reaction.

A comparison of the rate constants for dark bleaching (k_1_) and photobleaching (k_2_) for **22**–**26** derived from the slopes of the linearized plots indicates that k_2_ are slightly greater than k_1_, and generally the constants are of the same order of magnitude. Thus, the rate of relaxation of compounds **22**–**26** from the open-ring to the closed-ring form also does not depend on the number of the methylene groups in the spacer.

In addition, considerable variation was found for other important characteristics of photochromic compounds, such as photodegradation efficiency and light sensitivity defined as the ratio ∆D^max^/D^max^. Ongoing from **22** to **24**, the light sensitivity and the photostability of compounds increase. Further increase in the number of methylene groups in the chains from **24** to **26** leads to a decrease in both the light sensitivity and the stability to photodegradation (Figure 5).

It is known that the control spectral and luminescent properties of photochromic compounds is a promising trend in the field of photochromic technologies, owing to the exceptionally high sensitivity and luminescence response time. Meanwhile, it is known from the literature that the ammonium salt of spiropyran may act as an antitumor agent [18]. The detected luminescence behavior of the synthesized compound would enable visualization of the route and cellular location of a test compound, which is a promising trend in the design of anticancer drugs for humans. Therefore, we studied the spectral and luminescent properties of the photochromic spiropyran **22** in a THF solution at room temperature.

According to the experimental results, spirophotochrome **22** does not possess photoluminescence (PL) when it is in the closed spirocyclic form. However, UV irradiation of its THF solution at room temperature gives rise to PL in the red region of the visible spectrum with a peak at 645 nm (Figure 6). The measured PL quantum yield of **22** in merocyanine form was found to be low (ϕ_PL_ = 3.7 × 10^−4^).

The positions of peaks in the absorption spectra (Figure 6, curve 1) and in the PL excitation spectra (Figure 6, curve 2) of spiropyran **22** virtually coincide, indicating that the observed PL is due to the radiative transitions in the merocyanine form of **22**.

## 3. Materials and Methods

The detailed procedure of the synthesis and characterization of the products are given in Appendix A.

## 4. Conclusions

Thus, we synthesized new indoline-derived spiropyran salts containing functional substituents with different lengths of spacer between the indole and ammonium nitrogen atoms in the molecules, and studied their photochromic and luminescent properties. The results are that all synthesized compounds exhibit positive photochromism at room temperature. Study of the effect of structural factors on the spectral and kinetic characteristics of photochromic transformations of the synthesized spiropyrans demonstrated that the positions of the absorption bands of the closed and merocyanine forms of spiropyrans, and the rate constants of spiropyran relaxation from the open to closed form virtually do not depend on the length of the spacer between the indole and ammonium nitrogen atoms. Meanwhile, the light sensitivity and photostability of the compounds depends considerably on the number of methylene units. An increase in the number of methylene units induces first an increase and then a decrease in the light sensitivity and stability to photodegradation.

## Data Availability

Not applicable.

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
