# Peer review of "Synthesis, Photochromic and Luminescent Properties of Ammonium Salts of Spiropyrans"

_molecules, 2022, doi:10.3390/molecules27238492_

Round 1
Reviewer 1 Report
Khuzin and co-authors have synthesized several photochromic ammonium-functionalized spiropyrans. Photo-initiated reversible conversion of the spiropyrans has been investigated by UV-Vis absorption and fluorescence spectroscopy. And their quantum yields of photo-isomerization and spectral kinetic characteristics were measured. Notably, the authors found that the length of the linker between indole and ammonium nitrogen atoms significantly influenced the photosensitive and photostability of the spiropyrans, which provides a reference for developing the application of spiropyran derivates. I support its acceptance, however, some issues should be considered before publication.
1. Please rewrite the caption of Figure 1, which is very confusing. I suggest changing “l =0.1 cm” to “L = 0.1 cm”. Please also double-check Figure 4, which is Figure 4A?
2. How about the effect of the length of the linker between indole and ammonium nitrogen atoms on the reversibility and fatigue resistance of the spiropyrans?
3. How much are the conversion yields from the spiropyrans to merocyanines upon irradiation with UV light? Can they recover completely to spiropyrans upon irradiation with Visible light or kept in a dark condition?
4. Is it possible to detect the cis- and trans-isomers of merocyanine by 1H NMR? And how about their ratios?
5. What are the fluorescent quantum yields of the merocyanines?

Author Response
- Please rewrite the caption of Figure 1, which is very confusing. I suggest changing “l =0.1 cm” to “L = 0.1 cm”. Please also double-check Figure 4, which is Figure 4A?
Answer: The caption for Figure 1 has been corrected. Headings A and B have been added to figure 4, and the figure caption has been corrected.
- How about the effect of the length of the linker between indole and ammonium nitrogen atoms on the reversibility and fatigue resistance of the spiropyrans?
Answer: In the present work, studies to establish reversibility, fatigue strength, and the effect of the linker length on these factors were not carried out. According to the experimental data obtained in the study of the effect of the chain length on the degree of photodegradation of compounds 22–26, it can be predicted that the dependence of the fatigue strength limit will be similar, i.e. connections 22 and 26 will most likely have the lowest strength, and 23-25 will most likely have. It is also worth adding that all connections 22-26 demonstrated the ability to switch more than 10 times.
- How much are the conversion yields from the spiropyrans to merocyanines upon irradiation with UV light? Can they recover completely to spiropyrans upon irradiation with Visible light or kept in a dark condition?
Answer: The yields of conversion of spiropyrans to merocyanines under UV light measured with standard spectrophotometric method were 0.20 ± 0.03. The measured values are overestimated, so that we did not provide them in the work. This is due to the fact that the method does not take into account the change in proportion of light absorbed by merocyanine, when the absorption spectra of spiropyran and merocyanine overlap partially. We ordered additional equipment to measure the power of luminous flux in the visible and near-IR regions to find the conversion yields with a photokinetic method [M. Cipolloni, F. Ortica, L. Bougdid, C. Moustrou, U. Mazzucato, G. Favaro, New Thermally Irreversible and Fluorescent Photochromic Diarylethenes, J. Phys. Chem. A., 2008, V. 112, P. 4765-4771]. Current equipment and conventional actinometers, including Reinicke salts and rhodamine C, did not allow this to be done.
What about reduction of spiropyrans under irradiation with visible light or during storage in the dark, we can say the following. The compounds 22-26 in THF exhibited the ability to switch more than 10 times with a decrease in optical density intensity of no more than 10% under the standard conditions described in the work. The compounds, however, underwent the photochemical degradation, so that complete reduction is not observed.
- Is it possible to detect the cis- and trans-isomers of merocyanine by 1H NMR? And how about their ratios?
Answer: Cis-isomers of merocyanine are unstable and intermediate (their lifetime is a few femtoseconds) [Sheng Y. et al., J. Phys. Chem. B, 2004, 108, 16233-16243; Cottone G. et al., Chem. Phys. Lett., 2004, 388, 218-222; Maurel F. et al., J. Phys. Chem. A, 2006, 110, 4759-4771; Balasubramanian G. et al. Chem. Phys. Lett., 2012, 554, 60-66], and they transform immediately into stable trans-isomers in the solution after UV irradiation. Considering that NMR spectroscopy is a "slow" method in principle, it is impossible to detect these short-living cis-isomers of merocyanine with 1H NMR and, moreover, the cis-trans isomer ratios.
- What are the fluorescent quantum yields of the merocyanines?
Answer: In the present work, the spectral and luminescent properties of spirophotochrome 22 were studied. Using the standard procedure for estimating the relative quantum yields of PL (given in Supplementary information), the PL yield (ϕPL) of compound 22 in the merocyanine form was measured, equal to ϕPL = 3.7×10-4. The obtained low value of ϕPL makes it possible to classify compound 22 as weakly luminescent.
In the revised version of the manuscript, the sentence “However, UV irradiation of its THF solution at room temperature gives rise to fairly intense PL in the red region of the visible spectrum with a peak at 645 nm (Figure 6)” is replaced by “However, UV irradiation of its THF solution at room temperature gives rise to PL in the red region of the visible spectrum with a peak at 645 nm (Figure 6). The measured PL quantum yield of 22 in the merocyanine form was found to be low (ϕPL = 3.7·10-4)”.
Best regards,
on behalf of the co-authors,
Dr. Artur Khuzin
Reviewer 2 Report
The MS “Synthesis, photochromic and luminescent properties of ammonium salts of spiropyrans” authored by Khuzin and co-workers describes and interesting synthesis of new indoline spiropyrans aiming to expand the scope of applicability of spirophotochromic compounds with new light-controlled materials with a variety of properties including the effect of the length between the indole and ammonium nitrogen atoms on the spectral and the photochromic properties. Despite of that, as shown in the manuscript, a wide range of new salts derived from indoline spiropyrans were synthesized and showed an interesting capability of reversibly responding to UV radiation, being able to be measured with absorption and luminescence spectroscopy, and other spectral kinetic characteristics.
In summarize, minor revisions are required:
1 - The “3” of CDCl3 must be subscribed (i.e., CDCl3)
2 - In the Figure 1 is shown the absorption spectra of compound 22 in THF (C = 10-4 M, l = 0.1 cm. Is the length value 0.1 cm, correct?
3 - In the line 93, is mentioned that “the observed changes are reversible: molecules 22-26 cyclize again in the dark (dark bleaching) or under the action of visible light (photobleaching). This switching cycle can be repeated many times.” How long does it take to occur the change? In my opinion it should be in the content.
After these correction/inclusions, I will be happy in recommend the manuscript for publication in Molecules.
Author Response
- The “3” of CDCl3 must be subscribed (i.e., CDCl3)/
Answer: The typo has been corrected.
- In the Figure 1 is shown the absorption spectra of compound 22 in THF (C = 10-4M, l = 0.1 cm. Is the length value 0.1 cm, correct?
Answer: Yes, the optical path length (L) was 0.1 cm.
- In the line 93, is mentioned that “the observed changes are reversible: molecules 22-26cyclize again in the dark (dark bleaching) or under the action of visible light (photobleaching). This switching cycle can be repeated many times.” How long does it take to occur the change? In my opinion it should be in the content.
After these correction/inclusions, I will be happy in recommend the manuscript for publication in Molecules.
Answer: The complete photoinduced transition of spiropyrans 22-26 (10-4 M in THF) from closed to open form takes 11-18 seconds. The reverse transition is characterized by a longer duration: 180-190 seconds with dark discoloration and 100-170 seconds with visible light irradiation.
The revised version of the manuscript is supplemented with these details.
Best regards,
on behalf of the co-authors,
Dr. Artur Khuzin
Reviewer 3 Report
The paper by Khyzin et. al. describes the synthesis and photophysical exploration of ammonium appended spyropyrans. The work is well performed on the technical level. However, there couple of points that need to be fixed. First of all, a bit more analysis should be givem. Otherwise, the work is too technical.
Introduction should be extended.
Motivation for the synthesis of the selected cationic derivative should be given.
line 42 - CDCl3 - subscript
Is there any influence of the ammonium part on the photophysical properties?
Author Response
- Introduction should be extended.
Answer: The introduction has been updated. All new additions are marked by green.
- Motivation for the synthesis of the selected cationic derivative should be given.
Answer: The motivation for the choice of such a cationic derivative was the work published at the time of the beginning of our studies, which describes, using one example of spiropyran, photoswitchable cytotoxicity against Hek293 cancer cells (Chem. Commun 2011, 47, 11020). We have added this information to the introduction.
- 3. line 42 - CDCl3 – subscript
Answer: Typo fixed.
- Is there any influence of the ammonium part on the photophysical properties?
Answer: The nature and presence of various functional groups in spirophotochromic compounds, due to the high susceptibility of metastable states to external factors, significantly affects their photophysical properties. It can be assumed that the ammonium group will also affect the photophysical properties. However, such studies were not carried out in the present work. This is the subject of our future research.
Best regards,
on behalf of the co-authors,
Dr. Artur Khuzin
Round 2
Reviewer 1 Report
The author has improved the manuscript through some depth discussions, it is ready for publication in Molecules.